# Opportunities of Digital Infrastructures for Disease Management—Exemplified on COVID-19-Related Change in Diagnosis Counts for Diabetes-Related Eye Diseases

**DOI:** 10.3390/nu14102016

**Published:** 2022-05-11

**Authors:** Franziska Bathelt, Ines Reinecke, Yuan Peng, Elisa Henke, Jens Weidner, Martin Bartos, Robert Gött, Dagmar Waltemath, Katrin Engelmann, Peter EH Schwarz, Martin Sedlmayr

**Affiliations:** 1Institute for Medical Informatics and Biometry, Carl Gustav Carus Faculty of Medicine, Technische Universität Dresden, 01307 Dresden, Germany; franziska.bathelt@tu-dresden.de (F.B.); yuan.peng@tu-dresden.de (Y.P.); elisa.henke@tu-dresden.de (E.H.); jens.weidner@tu-dresden.de (J.W.); martin.sedlmayr@tu-dresden.de (M.S.); 2Department of Computer Science, Klinikum Chemnitz gGmbH, Flemmingstr. 2, 09116 Chemnitz, Germany; m.bartos@skc.de; 3Core Unit Datenintegrationszentrum, Universitätsmedizin Greifswald, Walther-Rathenau-Str. 48, 17475 Greifswald, Germany; robert.goett@med.uni-greifswald.de (R.G.); dagmar.waltemath}@med.uni-greifswald.de (D.W.); 4Department of Ophthalmology, Klinikum Chemnitz gGmbH, Flemmingstr. 2, 09116 Chemnitz, Germany; k.engelmann@skc.de; 5Department of Medicine, University of Dresden, Carl Gustav Carus, 01307 Dresden, Germany; peter.schwarz@ukdd.de

**Keywords:** diabetes, eye-disease, OMOP, COVID

## Abstract

**Background:** Retrospective research on real-world data provides the ability to gain evidence on specific topics especially when running across different sites in research networks. Those research networks have become increasingly relevant in recent years; not least due to the special situation caused by the COVID-19 pandemic. An important requirement for those networks is the data harmonization by ensuring the semantic interoperability. **Aims:** In this paper we demonstrate (1) how to facilitate digital infrastructures to run a retrospective study in a research network spread across university and non-university hospital sites; and (2) to answer a medical question on COVID-19 related change in diagnostic counts for diabetes-related eye diseases. **Materials and methods:** The study is retrospective and non-interventional and runs on medical case data documented in routine care at the participating sites. The technical infrastructure consists of the OMOP CDM and other OHDSI tools that is provided in a transferable format. An ETL process to transfer and harmonize the data to the OMOP CDM has been utilized. Cohort definitions for each year in observation have been created centrally and applied locally against medical case data of all participating sites and analyzed with descriptive statistics. **Results:** The analyses showed an expectable drop of the total number of diagnoses and the diagnoses for diabetes in general; whereas the number of diagnoses for diabetes-related eye diseases surprisingly decreased stronger compared to non-eye diseases. Differences in relative changes of diagnoses counts between sites show an urgent need to process multi-centric studies rather than single-site studies to reduce bias in the data. **Conclusions:** This study has demonstrated the ability to utilize an existing portable and standardized infrastructure and ETL process from a university hospital setting and transfer it to non-university sites. From a medical perspective further activity is needed to evaluate data quality of the utilized real-world data documented in routine care and to investigate its eligibility of this data for research.

## 1. Introduction

Cross-national and intersectoral research on real-world data (RWD) offers the opportunity for unbiased generation of new knowledge and its further usage in the future. On the one hand, this includes the retrospective recognition of bottlenecks in healthcare processes and the definition of countermeasures that can be utilized directly for a continuous process monitoring. On the other hand, evidence for diagnostics and therapeutic measures can be obtained and thus the results can support personalized care in a prospective setting (e.g., [1]).

In this context, the development of a research network for healthcare across countries is being steadily advanced at the European level. With EHDEN [2], a European, open science collaboration operating on anonymized RWD to gain evidence on specific topics facilitating standardized methodologies was set up in 2018, and is growing steadily. Recently the European Medical Agency started the initiative “Data Analysis and Real World Interrogation Network (DARWIN EU^®^)”, which focuses on “… timely and reliable evidence on the use, safety and effectiveness of medicines for human use …” [3]. Both initiatives, EHDEN and DARWIN, are based on the Observational Medical Outcomes Partnership (OMOP) common data model (CDM), which is provided by Observational Health Data Sciences and Informatics (OHDSI). With that standardized data model, semantic interoperability is ensured across countries. As shown by Reinecke et al. [4], the OMOP CDM is increasingly used worldwide for retrospective research, as it fosters research at large scale and ensures reliability of the research results.

In Germany, a first step towards the participation in such research networks was taken in 2018 by the initiation of the German Medical Informatics Initiative (MII) [5]. The MII realizes the semantical interoperability among all university hospitals in Germany by the development and usage of a core dataset (CDS) based on Health Level Seven International Fast Healthcare Interoperability Resources (HL7 FHIR) [6] (p. 7).

Based on this MII CDS, the center for medical informatics in Dresden (ZMI) designed and implemented an extract, transform and load (ETL) process to move data to the OMOP CDM [7]. The successful deployment of this ETL process at ten university hospitals has been accomplished by the Medical Informatics in Research and Care in University Medicine (MIRACUM) team, which is one of the MII consortia [8].

The digital progress hub Medical Informatics Hub in Saxony (MiHUBx), launched in 2021 is going to overcome former limitations of the MII and thus extends existing research infrastructures to non-university healthcare service providers. The interoperability of non-university healthcare service providers with the MII will be supported by providing a regional core infrastructure based on the specifications of the MII.

MiHUBx is required to demonstrate benefits of healthcare data exchange for the healthcare sector by addressing medical use cases. The use case “Ophthalmology meets Diabetology” aims to provide intersectoral IT-based diagnostic and therapy decision support based on RWD. A fundamental requirement to support this use case is the data interoperability between university and non-university healthcare providers.

We focus on the high prevalence diseases diabetes mellitus type 1 and type 2, as well as diabetic eye diseases—diabetic retinopathy and diabetic macular edema [9]. This group of patients has been highly affected by the outbreak of the SARS-CoV-2 pandemic, as diabetic patients have been at high risk of increased severity or of adverse outcomes [10]. This led to a fear in patients of being exposed to the virus and, at the same time, a worsening progress of the diabetic related eye diseases [11]. Especially during lockdown, actions ongoing or planned treatments have been skipped or postponed reinforcing existing patient fears [12,13].

In this paper, we present opportunities created by digital infrastructures while ensuring the interoperability of data for the cross-site use of data for research on RWD using the example of diabetes mellitus type 1 and 2, as well as the above-mentioned diabetic eye diseases. In this context we focus on the following two aspects:(1)*Technical view*: The paper focuses on the ability to establish a concept and infrastructure that uses OMOP CDM across 3 different sites within MIRACUM and the MiHUBx project. We aim to answer the question whether OMOP can be successfully used by both university and non-university healthcare providers to support feasibility requests required to participate in multi-centric studies.(2)*Medical view:* We run a multi-centric study on the COVID-19-related change in diagnosis counts for diabetes-related eye diseases, based on the provided infrastructure components in (1). This paper aims to answer whether the number of diagnoses in Germany for diabetes mellitus type 1/2; for diabetic retinopathy and for diabetic macular edema changed in pandemic times (January 2020–December 2021) compared to the period before the SARS-CoV-2 pandemic (January 2018–December 2019).

## 2. Materials and Methods

### 2.1. Setting

#### 2.1.1. Infrastructural Concept

The analysis is carried out within the scope of the MII at the University Medicine Greifswald, Dresden university hospital, and at the non-university hospital Chemnitz. The required technological platform—consisting of ETL processes and OHDSI tools that were provided by Dresden as docker-containers based on Gruhl et al. [14] and Henke et al. [7]. These docker-containers were deployed at each participating site. The technical architecture is shown in Figure 1. Regular meetings between all participants offered room for discussion and technical questions. It ensures tracking of progress and early detection of obstacles to overcome.

#### 2.1.2. ETL-Processes and OHDSI Tools

The data transfer of the medical case data of the participating sites to the OMOP CDM is an essential step for collaborative work and research that needs to be done first. As described in Chapter 6 in the Book of OHDSI [15], it consists of an ETL process. It harmonizes the original data of the participating sites that exists in different and appropriate formats and structures. A central component of the ETL process is the mapping of national data formats to standardized vocabularies, such as SNOMED-CT or LOINC, forced by the OMOP CDM and the OHDSI tools. The standardized vocabularies delivered by the OHDSI service ATHENA ensure the semantic interoperability of medical terms and clinical facts and thus pave the way for standardized analyses across multiple sites.

The OHDSI tool stack used in this paper consists of the OMOP CDM database version 5.3.1 and the web application ATLAS with the WebAPI version 2.10.1. The OMOP database was delivered on PosgreSQL version 13.1. The ATLAS web application is the common user interface to design, implement and run analyses against data available in OMOP CDM. ATLAS allows users to connect to one or more OMOP databases, it provides search options, cohort definition functionality and a lot more data analytics opportunities, as described in detail in Chapters 7–13 in the Book of OHDSI [15].

#### 2.1.3. Technical Data Acquisition and Cohort Definition

The medical study is a retrospective, non-interventional study that uses medical case data documented in routine care at the participating sites. For this purpose, the already established data structures, i.e., HL7 FHIR basic modules of the MII core dataset and an ETL process that transfers data from FHIR to OMOP CDM [7], were used at the university site in Dresden.

For the participating non-university site Chemnitz, medical case data documented in routine care was used which corresponds the specification of German claims data according to §21 KHEntgG (engl. hospital fees law) and are stored as comma separated files (CSV) files [16]. The transfer to OMOP CDM has been done based on the P21 to OMOP CDM ETL-process. The relevant mappings of this ETL process are identical to the ETL process from FHIR to OMOP used for the university site [17]. All data necessary for this study have been transformed into local OMOP CDM database instances at each site.

Four cohort categories (Table 1) were defined via OHDSI ATLAS to carry out the queries on the OMOP CDM database, each for one of the four years under consideration (16 cohorts in total). The cohort definition was done by computer scientists (see Appendix B) and reviewed by medical staff. The centrally specified cohorts were exported as JavaScript Object Notation (JSON) files to ensure the usage of the same cohort definitions for all participating sites. This approach allowed the team to distribute the cohorts to all sites and avoid possible errors introduced by human interaction. After the import of the cohort definitions to the ATLAS instance at each participating site, the processing by the respective DIC was carried out. Finally, each site reported the results for all 16 cohorts back to the ZMI.

In accordance with the comprehensive data protection concept of the MII in conjunction with the local data protection concept of the data integration center in Dresden, all data remained at the location data were generated at, and thus data did not leave the territory of the participating sites.

### 2.2. Study Design

The medical view has been conducted as a retrospective observational study evaluating routine data.

#### 2.2.1. Eligibility Criteria

To answer the primary and secondary questions, all inpatient visits between 2018 and 2021 were included. The study requires diagnoses coded in ICD-10-GM. The cohorts were defined, as shown in Table 1.

#### 2.2.2. Sample Size

The sample size refers to all inpatient medical cases performed at participating hospitals in 2018–2021. Over all sites, this comprises 803,467 unique patients.

#### 2.2.3. Ethics

An ethics vote was not required for this study because no patients were directly involved, and only indirect patient data were used for analysis.

### 2.3. Primary and Secondary Outcome

Quantitative changes in the number of diagnoses at pandemic times compared with pre-pandemic times were checked. For this, data from 2018 and 2019 were used as baseline values and compared with data for diagnoses documented in 2020 and 2021 for each condition.

### 2.4. Data Analyses

Data analyses and visualization were realized using python via Python and Jupyter notebook (see Appendix A). The implementation consists of the calculation of the relative changes within a cohort category (I–IV) (a) between each year and (b) between the year category (pre-pandemic: 2018 and 2019, and pandemic: 2020 and 2021). The year-based results have finally been used for the quantitative comparison between the cohort categories.

## 3. Results

### 3.1. Technical View

The provision and integration of docker-containers at sites with an established research infrastructure did not raise any technical problems. At sites without an existing infrastructure and missing expertise regarding docker, the implementation caused some problems mainly regarding the understanding of error-messages. Thus, for those sites, a virtual machine has been developed and deployed, which already consisted of integrated docker-containers. For ease of usage, shell scripts calling the right commands were included into the virtual machine (see Figure 2). The use of the virtual machine did not cause any technical problems.

### 3.2. Medical View

#### 3.2.1. Feasibility Results

All participating sites applied the centrally defined cohorts and reported the following numbers (see Table 2).

#### 3.2.2. Change in Diagnosis Numbers

During the COVID-19-pandemic (2020–2021) total diagnoses dropped by 17.76%, compared to pre-pandemic times (2018–2019), across all participating sites. The number of diagnoses for non-diabetes patients (−17.59%) and diabetes patient (−19.40%) decreased roughly to the same extent (Figure 3a).

However, the number of diagnoses for diabetes-related eye diseases declined more sharply (Figure 3b). For cohort category II (diabetes + retinopathy) the numbers dropped by 21.94% during the pandemic compared to pre-pandemic times. For cohort category III (diabetes + makula edema) the numbers decreased by 24.25% (Table 3).

Diagnostic numbers were found to vary between sites (Table 2). While the numbers for cohort category III declined sharply at two sites, records from the third site showed consistent diagnosis numbers. A slight increase in numbers was identified for cohort category II at site 2, whereas numbers for this cohort category dropped for the other sites. In categories I and IV, there was a decrease in diagnostic numbers at all sites, with the relative decrease in diagnostic numbers varying between sites (Table 3).

## 4. Discussion

From a medical perspective, the decline in diagnoses is consistent with the studies presented at the outset. However, the difference between the relative numbers of diabetes and diabetes-related eye diseases is a bit surprising. One reason for this could relate to the documentation of pre-existing conditions in hospitalized COVID patients requiring treatment during their stay. For example, diabetes may have been documented more frequently compared with diabetes-related eye diseases. Another possibility is a dramatic increase of diabetes patients despite the pandemic, as indicated in previous studies [18,19], and thus the decline in numbers is less than for diabetes-related eye diseases. Further studies with additional data elements should be conducted to investigate causality.

From a technical perspective, the multi-centric study worked well at all sites. The cohort distribution required manual effort that could be reduced by applying a cross-site data sharing approach [20] in the future. The tool stack, based on standardized terminologies used in this study, allows easy and time saving integration of new sites and can be used for other research questions in the future. The infrastructure setup for this study based on the OMOP CDM enables German sites to participate on international studies executed by large research networks in the near future.

However, the availability of structured routine data of a high quality is essential, and may impact research results if missing. Close cooperation between computer scientists and medical staff is crucial and necessary to identify gaps and issues in the routine data and to define new features and function of the existing infrastructure to run further retrospective studies in the same setting. Therefore, we recommend working as an interdisciplinary team from the very beginning until the end of a project, covering technical and medical topics during the design and development of digital infrastructures, the formal definition of relevant cohorts within these infrastructures and the execution of multi-centric studies. This approach helps both expert groups, medical staff, and computer scientists, to broaden their knowledge, to gain new perspectives and to ensure overall project success together. Routine data quality and the impact of using routine data that are mainly documented for claim purposes, need further investigation to ensure result interpretation from the medical perspective is feasible for future studies with more complex research questions, and to add value in terms of diagnoses and therapy improvements.

The presented work is limited since only three different sites participated, and only one of the sites was a non-university hospital. The data used so far include inpatient visits only, no resident doctors or other outpatient providers have been integrated. The study is also limited in terms of utilized terminologies, since only ICD-10GM diagnoses have been used to answer the research question. Thus, future work is needed to increase the number of participating sites. Additionally, outpatient data need to be harmonized for the secondary usage in research on the infrastructure platform, as intended by the digital progress hub MiHUBx in a next step.

## 5. Conclusions

With the technical setup, a multi-centric study between university and non-university hospitals based on local routine data became possible. The OMOP CDM and the OHDSI tools allow a participation at international studies in the near future. That can strengthen existing networks such as the “Diabetic Retinopathy Clinical Research Network” [21]. Although we restricted the cohort definitions to ICD-10GM coded diagnoses, a translation to SNOMED-CT is already available based on the OHDSI vocabulary service ATHENA, that includes mappings for 98.8% ICD-10GM codes to SNOMED-CT [22].

From a medical perspective, the availability of structured data builds the base for additional tools, such as machine learning algorithms (e.g., [23]) and decision support systems (e.g., [24,25]), which result in suggestions for diagnostic and therapy options, especially, but not limited to, diabetes-related eye-diseases.

## Figures and Tables

**Figure 1 nutrients-14-02016-f001:**
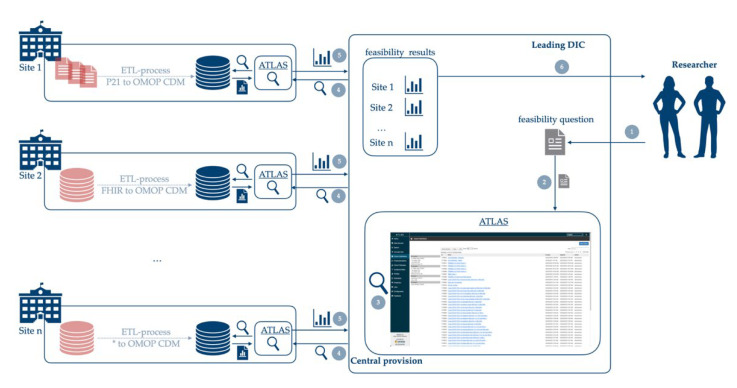
Technical architecture: Researcher poses a feasibility question to a leading data integration center (DIC) (1). A medical informatics specialist formalizes the feasibility question (2). Then the specialist builds and exports corresponding cohorts in ATLAS (3) and sends them to participating sites (4). The participating sites import the cohort and generate results, which are then sent back to the leading DIC (5). The leading DIC combines the results, pseudonymizes the sites and provide this result to the requesting researcher (6).

**Figure 2 nutrients-14-02016-f002:**
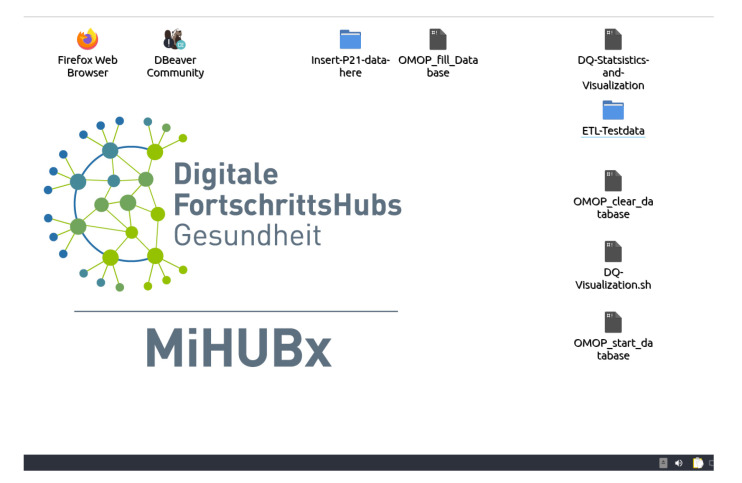
Screenshot of the virtual machine. Black icons represent the included shell scripts.

**Figure 3 nutrients-14-02016-f003:**
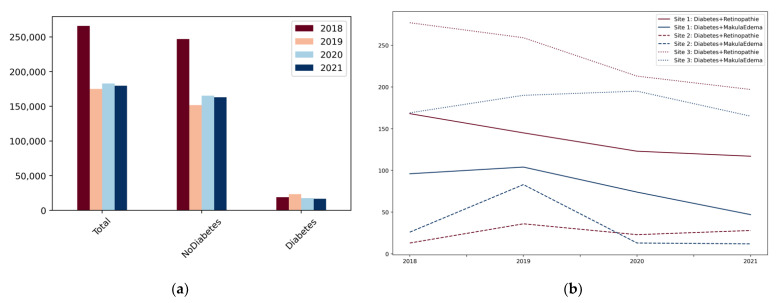
(**a**) Resulting numbers for the total amount of diagnoses, diagnoses other than diabetes and diabetes diagnoses over the years 2018–2021 and over both sites; (**b**) resulting numbers for diagnoses of diabetic related eye diseases.

**Table 1 nutrients-14-02016-t001:** Cohort definitions.

Cohort Category	Cohort ID	Year	Diagnosis
I		*Diagnosis of Diabetes type 1 or type 2 in corresponding year*	
	1	2018	ICD-Code E10. × or E11. ×
	2	2019	ICD-Code E10. × or E11. ×
	3	2020	ICD-Code E10. × or E11. ×
	4	2021	ICD-Code E10. × or E11. ×
II		*Diagnosis of Diabetes type 1 or type 2 with Diagnosis Retinopathia diabetica in corresponding year*	
	5	2018	(ICD-Code E10. × or E11. ×) AND (ICD-Secondary-Code H36. ×)
	6	2019	(ICD-Code E10. × or E11. ×) AND (ICD-Secondary-Code H36. ×)
	7	2020	(ICD-Code E10. × or E11. ×) AND (ICD-Secondary-Code H36. ×)
	8	2021	(ICD-Code E10. × or E11. ×) AND (ICD-Secondary-Code H36. ×)
III		*Diagnosis of Diabetes type 1 or type 2 with Diagnosis makula edema in corresponding year*	
	9	2018	(ICD-Code E10. × or E11. ×) AND (ICD-Code H35. ×)
	10	2019	(ICD-Code E10. × or E11. ×) AND (ICD-Code H35. ×)
	11	2020	(ICD-Code E10. × or E11. ×) AND (ICD-Code H35. ×)
	12	2021	(ICD-Code E10. × or E11. ×) AND (ICD-Code H35. ×)
IV		*Other Diagnoses than defined in 1–12 in corresponding year*	
	13	2018	NOT (ICD-Code E10. × or E11. ×)
	14	2019	NOT (ICD-Code E10. × or E11. ×)
	15	2020	NOT (ICD-Code E10. × or E11. ×)
	16	2021	NOT (ICD-Code E10. × or E11. ×)

**Table 2 nutrients-14-02016-t002:** Feasibility results.

Cohort Category	Cohort ID	Year	Site 1	Site 2	Site 3
	*diagnosis of diabetes type 1 or type 2 in corresponding year*				
	1	2018	6.168	4.073	8.877
I	2	2019	6.272	8.177	8.946
	3	2020	6.024	3.670	7.870
	4	2021	5.814	3.763	7.123
	*diagnosis of diabetes type 1 or type 2 with diagnosis retinopathia* *diabetica in corresponding year*				
	5	2018	168	13	277
II	6	2019	145	36	259
	7	2020	123	23	213
	8	2021	117	28	197
	*diagnosis of diabetes type 1 or type 2 with diagnosis makula edema in corresponding year*				
	9	2018	96	26	169
III	10	2019	104	83	190
	11	2020	74	13	195
	12	2021	47	12	165
	*other diagnoses than defined in 1–12 in corresponding year*				
	13	2018	42.759	151.239	52.771
IV	14	2019	43.446	56.114	51.954
	15	2020	41.223	78.844	45.213
	16	2021	40.084	78.566	43.613

**Table 3 nutrients-14-02016-t003:** Quantitative changes in diagnosis numbers between pre-pandemic (year 2018 + 2019) and pandemic (year 2020 + 2021) by cohort category and site.

Cohort Category	Site 1	Site 2	Site 3	Total
*changes between year groups for diagnosis of diabetes type 1 or 2*				
I	−4.84%	−39.32%	−15.88%	**−19.40%**
*changes between year groups for diagnosis of diabetes type 1 or 2 with diagnosis retinopathia diabetica*				
II	−23.32%	+4.08%	−23.51%	**−21.94%**
*changes between year groups for diagnosis of diabetes type 1 or 2 with diagnosis macula edema*				
III	−39.50%	−77.06%	+0.28%	**−24.25%**
*changes between year groups for other diagnoses than defined in 1–12*				
IV	−4.79%	−24.12%	−15.18%	**−17.59%**

## Data Availability

Cohort definitions based on OMOP CDM are provided in the Appendix B. No further data can be provided due to legal restrictions.

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
