# Peer review of "Opportunities of Digital Infrastructures for Disease Management—Exemplified on COVID-19-Related Change in Diagnosis Counts for Diabetes-Related Eye Diseases"

_nutrients, 2022, doi:10.3390/nu14102016_

Round 1

Reviewer 1 Report

Very current topic, look at these points to improve the manuscript:

  • the whole paper is very confusing and hard to read, it should be revised and made easier to read
  • the aim of the paper is not well reported. Please revise.
  • Add more details for table 3.
  • Discussion section is very short. Revise and improve.
  • Lines 268-270: "A close cooperation between computer scientists and medical staff is crucial and needed to identify gaps and issues in the routine data and to define..." What do authors proposed to increase this cooperation?

Author Response

Dear Reviewer,

thanks a lot for your valuable feedback. We really appreciate it.

We have revised our paper according to your feedback as follows:

1 - we revised the aim of the paper to ensure readability is improved and no confusion is generated for the reader

2 - we revised our results section, in particular we added more information to Table 3, corrected the name and details that have been wrong by accident before, we also added some more details to the results section 

3 - we revised our discussion: We added a proposal to increase the cooperation, we added missing limitation and future work as suggested

If needed we can also provide our revised version of the paper with "track changes=on" that would allow you to follow our changes in detail, but unfortunately I am not allowed to upload more than one word document. 

Sincerely yours 

Reviewer 2 Report

Dear Authors,

Thank you for the opportunity to review the original article entitled „Opportunities of digital infrastructures for disease management - Exemplified on Covid19-related change in diagnosis counts for diabetes-related eye diseases” which addresses the development of a network to facilitate the implementation of an integrated information system to improve the processing of databases from multiple centers. This is an important point because it improves the accuracy of data collected and it also gives the possibility to achieve representative cohorts to be able to use the data for the purpose of a wide range of research.

The usefulness of this system is demonstrated by the researchers of this study, by using a newly developed digital infrastructure to obtain real-life data. Thus, it was noticed a decrease in the addressability of diabetic patients, and especially those with diabetic retinopathy, during the SARS-COV2 pandemic waves.

The study is correctly designed and the research methodology is in line with the proposed objectives, but my medical training limits my ability to appreciate it in terms of technical details specific to computer science The methods used in this research are well described and provide sufficient details to be understood.

The results are appropriately interpreted and respond to the aim of the study.

The discussions address the findings of the research, especially from the point of view of the development of the computer network necessary for the acquisition and processing of primary data from the patients. Discussions could be developed if the authors could answer the following questions

  1. Could they identify database management systems developed in other European countries?
  2. What would be the limitations of this digital infrastructure? Does it use only the data obtained by the ICD coding system or other data can be entered?

Thank you for your esteemed efforts in increasing our collective knowledge.

Sincerely yours

Author Response

Dear Reviewer,

thanks a lot for your valuable feedback. We really appreciate it.

Thanks for the ideas on further discussion topics (1) other DBMS systems in Europe and (2) data other than ICD codes. We added information on data other than ICD codes to our discussion as well as limitation and future work. 

If needed we can also provide our revised version of the paper with "track changes=on" that would allow you to follow our changes in detail, but unfortunately I am not allowed to upload more than one word document. 

Sincerely yours 

Round 2

Reviewer 1 Report

good